# Quantification of Carotid Intraplaque Hemorrhage: Comparison between Manual Segmentation and Semi-Automatic Segmentation on Magnetization-Prepared Rapid Acquisition with Gradient-Echo Sequences

**DOI:** 10.3390/diagnostics9040184

**Published:** 2019-11-11

**Authors:** Young Ju Song, Hyo Sung Kwak, Gyung Ho Chung, Seongil Jo

**Affiliations:** 1Department of Radiology of Chonbuk National University Hospital, Jeon-ju 54907, Korea; twinklinglena@gmail.com; 2Radiology and Research Institute of Clinical Medicine of Chonbuk National University-Biomedical Research Institute of Chonbuk National University Hospital, Jeon-ju 54907, Korea; chunggh@jbnu.ac.kr; 3Department of Statistics (Institute of Applied Statistics), Chonbuk National University, Jeon-ju 54907, Korea; statjs@jbnu.ac.kr

**Keywords:** carotid intraplaque hemorrhage, carotid atherosclerosis, semi-automatic quantification, MPRAGE

## Abstract

Purpose: Carotid intraplaque hemorrhage (IPH) increases risk of territorial cerebral ischemic events, but different sequences or criteria have been used to diagnose or quantify carotid IPH. The purpose of this study was to compare manual segmentation and semi-automatic segmentation for quantification of carotid IPH on magnetization-prepared rapid acquisition with gradient-echo (MPRAGE) sequences. Methods: Forty patients with 16–79% carotid stenosis and IPH on MPRAGE sequences were reviewed by two trained radiologists with more than five years of specialized experience in carotid plaque characterization with carotid plaque MRI. Initially, the radiologists manually viewed the IPH based on the MPRAGE sequence. IPH volume was then measured by three different semi-automatic methods, with high signal intensity 150%, 175%, and 200%, respectively, above that of adjacent muscle on the MPRAGE sequence. Agreement on measurements between manual segmentation and semi-automatic segmentation was assessed using the intraclass correlation coefficient (ICC). Results: There was near-perfect agreement between manual segmentation and the 150% and 175% criteria for semi-automatic segmentation in quantification of IPH volume. The ICC of each semi-automatic segmentation were as follows: 150% criteria: 0.861, 175% criteria: 0.809, 200% criteria: 0.491. The ICC value of manual vs. 150% criteria and manual vs. 175% criteria were significantly better than the manual vs. 200% criteria (*p* < 0.001). Conclusions: The ICC of 150% and 175% criteria for semi-automatic segmentation are more reliable for quantification of IPH volume. Semi-automatic classification tools may be beneficial in large-scale multicenter studies by reducing image analysis time and avoiding bias between human reviewers.

## 1. Introduction

Intraplaque hemorrhage (IPH) is considered to have an important role in the progression of atherosclerosis [1,2]. It has been known to be a critical marker of plaque progression and destabilization created by proinflammatory response and cholesterol-rich erythrocyte membrane accumulation [1,3,4]. IPH can also lead to a higher risk of cerebrovascular events [5]. Therefore, it is important to detect and quantify IPH in order to prevent atherosclerotic diseases such as territorial cerebral ischemic events.

Magnetic resonance (MR) sequences of carotid plaques are widely used for detecting and quantifying carotid IPH, using histologic analysis as the gold standard [6,7,8,9]. T1-weighted MR sequences have been commonly used to detect IPH, which shows as a high signal intensity on T1-weighted images due to T1 shortening. This is caused by hemoglobin degrading into methemoglobin in the intraplaque hemorrhage. T1-weighted sequences, such as two-dimensional fast spin-echo, three-dimensional time-of-flight (TOF), and three-dimensional magnetization-prepared rapid acquisition gradient-echo (MPRAGE) sequences, are used to detect and quantify IPH [6,9,10,11,12]. Our study used an MPRAGE sequence due to its superior diagnostic capability for detecting and quantifying IPH compared to fast spin-echo and TOF sequences [12]. MPRAGE imaging expedites signal suppression from background tissues by using a nonselective inversion pulse and spectrally selective water excitation or fat suppression [13,14].

MR-depicted IPH is mostly defined as high plaque signal intensity exceeding the intensity of adjacent muscle [12,15]. In previous studies, manual segmentation of plaque components was performed by delineating the lumen and outer wall for hyperintense areas on T1-weighted sequences [15,16]. We thought that manual segmentation for analysis of the plaque component can be time-consuming and contains bias between human reviewers due to different levels of experience especially in large-scale multicenter studies. To solve these problems related to manual segmentation, automatic segmentation algorithms have been used to assess IPH volume [17]. According to recent studies, automated plaque component analysis software can be as accurate as manual analysis [18,19]. Automatic analysis commonly defines IPH as high signal intensity on MPRAGE imaging that is greater than 200% of the intensity of adjacent muscle in at least two consecutive slices. This method has been histologically validated using a 3.0 T MRI machine [12]. However, different sequences or criteria were used to diagnose and quantify carotid IPH.

This study compared manual segmentation and semi-automatic segmentation for quantification of carotid IPH on widely available magnetization-prepared rapid acquisition with gradient-echo (MPRAGE) sequences. We aimed to achieve an optimized hyperintensity threshold for semi-automatic carotid IPH quantification on MPRAGE sequences.

## 2. Materials and Methods

### 2.1. Study Subjects

This study was approved by the institutional review board of our institution (Chonbuk National University; 2016-07-037), and informed consent was waived by our Institutional Review Board. We included 40 consecutive patients with 16–79% carotid stenosis and IPH on MPRAGE sequences that underwent carotid MR imaging to evaluate plaque components between January 2016 and October 2018.

### 2.2. Carotid MRI Protocol

Carotid plaque MR examination was performed on an Achieva 3.0-T scanner (Philips Medical System, Best, the Netherlands) with a 16-channel head coil. The carotid artery with the dominant plaque was used to center the MR scan, and this image was used in the image analysis. Our protocol for carotid plaque MR imaging included carotid MPRAGE imaging, and sequences were centered at the bifurcation of the carotid artery with plaque. The MPRAGE sequence was as follows: TR/TE = 13.2 ms/3.2 ms, flip angle = 15°, in plan spatial resolution = 0.63 mm × 0.63 mm, reconstructed resolution = 0.31 mm × 0.31 mm, slice thickness = 1 mm, number of excitation = 2, TI = 304 ms, TR with respect to the nonselective inversion = 568 ms, acquisition time = 3 min 50 s. Fat suppression was achieved by water selective excitation.

### 2.3. Image Analysis

Manual review: Carotid plaques were defined when the carotid artery’s wall thickness was greater than 2 mm in at least two consecutive slices on MPRAGE imaging. Interpretation of MPRAGE images for IPH detection of carotid plaque was done using plaque analysis software (MRI-PlaqueView, VP Diagnostics, Seattle, WA, USA). Data was then analyzed by researchers who were trained in the evaluation of carotid plaque MRI and blinded to study goals. Initially, the radiologists manually read the IPH based on the MPRAGE sequence but were blinded to all clinical variables. IPH segmentation was performed with a piecewise smooth regional level set method (Figure 1).

Semi-automatic review: Semi-automatic analysis of IPH volume was performed by one reader using the three-dimensional image solution program (Rapidia, Infinite, Seoul, Korea). IPH on MPRAGE images was defined as the presence of hyperintense plaque of signal intensity of the adjacent muscle for at least two consecutive sections. Initially, we measured the signal intensity of three points in the sternocleidomastoid muscle (SCM) and calculated the mean signal intensity of SCM (Figure 2a). IPH detection and volume were measured by the area that showed signal intensity of >150%, >175%, and >200% of the mean intensity of the SCM (Figure 2b–d). We analyzed the total IPH volume, maximal axial IPH volume, and length of IPH with each signal intensity on MPRAGE images.

### 2.4. Statistical Analysis

All statistical analyses were performed in the R language ver. 3.3.2 (R Foundation for Statistical Computing, Vienna, Austria). Agreement on measurements between manual segmentation and the three criteria of semi-automatic segmentation was assessed using the intraclass correlation coefficient (ICC). The ICC value was interpreted as poor (0), slight (0.01–0.20), fair (0.21–0.40), moderate (0.41–0.60), substantial (0.61–0.80), and almost perfect (0.81–1.00). Comparison of ICC value between manual and semi-automatic segmentation was performed with the Fisher’s test with Bonferroni correction. Statistical significance was defined as a *p* value less than 0.05.

## 3. Results

The ICC between manual segmentation and the three criteria of semi-automatic segmentation for quantification of IPH volume is shown in Table 1. The ICC of each semi-automatic segmentation compared with manual segmentation was as follows: 150% criteria = 0.861, 175% criteria = 0.809, 200% criteria = 0.491. There was almost perfect agreement between manual segmentation and the 150% criteria and 175% criteria of semi-automatic segmentation for quantification of IPH volume (Table 1 and Table 2). The ICC between manual and the 200% criteria appeared to have moderate agreement.

The ICC value of manual vs. 150% criteria and manual vs. 175% criteria were significantly better than that of manual vs. 200% criteria (*p* < 0.001) (Figure 3).

## 4. Discussion

In previous studies, IPH was manually characterized as high signal intensity by visual examination or using an arbitrary threshold of 1.5 to 2.0 compared to the signal intensity of adjacent muscle or normal arterial wall on MPRAGE sequences [12,20,21,22,23]. But manual IPH detection can be time-consuming and has lower repeatability and poor interobserver reproducibility compared to semi-automatic methods due to less distinguished outlines of IPH and subjective criteria. Ota et al. [12] manually identified IPH using MPRAGE sequences on 20 patients that were planned for carotid endarterectomy and assessed the performance based on histologic analysis. Liu et al. [24] used the semi-automatic median technique and achieved better IPH area correlation with histology than Ota et al. [12]. Touze et al. [25] reported that the intra-observer and inter-observer reproducibility on manual IPH area quantification were suboptimal (ICC = 0.70 and 0.60, respectively). As such, semi-automatic IPH segmentation can provide more accuracy and efficiency compared to manual segmentation, especially in large-scale multicenter studies.

Carotid IPH is known to have an association with cerebral ischemic events [5,26,27,28]. A meta-analysis demonstrated by Hosseini et al. [28] featured a strong association between the presence of carotid IPH and recurrent cerebral ischemic events in symptomatic carotid artery stenosis (OR 12.1, 95% CI 5.5–27.1, *p* < 0.001), which suggests IPH is a strong predictor of future cerebrovascular events in patients with carotid artery stenosis. Another meta-analysis according to Saam T et al. [5], showed that the presence of IPH was related to a ~6-fold higher risk for events (hazard ratio 5.69) in 8 studies with 689 patients, and the annualized event rate in subjects with detectable IPH was 17.7% compared with 2.4% in patients without IPH. Furthermore, previous studies suggest that carotid IPH may lead to macrophage accumulation and fibrous cap degradation, which stimulates plaque progression [29,30,31]. Thereby, early detection of IPH is of great importance for preventing future cerebral ischemic events and plaque progression. This study facilitates the detection and quantification of IPH that are prone to prompt strokes in the future.

Various MR sequence techniques have been used to reliably identify carotid IPH in T1-weighted images, such as fast spin echo, time-of-flight (TOF) MR angiography (MRA), and MPRAGE sequences. According to Saito et al. [32], carotid IPH signal differences showed similar accuracies between fast spin echo, TOF by 1.5-T MR, and MPRAGE sequences for identifying plaque components, based on histopathological confirmation. In contrast, Ota et al. [12] reported that MPRAGE sequences demonstrated superior diagnostic capability for detecting and quantifying IPH compared to fast spin echo and TOF sequences, using histologic analysis as the standard of reference. Although small IPHs or heavily calcified IPHs were excluded, MPRAGE sequences showed higher sensitivity (80%) and specificity (97%) than TOF imaging. Consequently, our study used MPRAGE sequences for the MR protocol to detect and quantify early carotid IPH, which can take up to 3–4 min to acquire.

Visual assessment of atherosclerotic carotid plaque has been used to determine plaque components with adjacent neck muscle as a standard reference. On TOF source images, IPH was defined as a presence of high signal intensity within the carotid plaque that is greater than 150% of the signal intensity of the adjacent neck muscle [10]. In MPRAGE imaging, plaque is thought to be “MPRAGE positive” if the signal intensity is greater than 200% of the intensity of the adjacent sternocleidomastoid muscle in at least two consecutive slices [33]. This method has been previously validated with histologic confirmation using a 3.0 T MRI [12]. However, there is no known standard reference to define IPH on MPRAGE imaging.

A recent study by Liu et al. [24] evaluated semi-automatic IPH characterization using three tissue references: (1) the mean signal intensity of sternocleidomastoid muscle (SCM); (2) the mean signal intensity of adjacent muscle; (3) the median signal intensity within the 4 cm circular region of interest (ROI) which has been used as a reference in previous studies for signal normalization in automatic plaque segmentation. As a result, using an optimized intensity threshold of 1.6 times the adjacent muscle showed good performance on IPH detection and quantification in MPRAGE sequences whereas using the SCM intensity as a reference was not recommended without coil sensitivity correction when surface coils were used. Similarly, our study focused on comparing manual segmentation and semi-automatic segmentation of IPH using the criteria of high signal intensity on MPRAGE imaging that is greater than 150%, 175%, and 200% of the intensity of adjacent muscle but included a larger number of patients. Our result shows that the ICC of the 150% and 175% criteria were significantly better than that of the 200% criteria, and this resembles Liu et al.’s [24] optimized intensity threshold of 1.6 times the adjacent muscle. To our knowledge, not a lot of studies are known for determining the optimal objective criteria for IPH detection and quantification using MPRAGE images. Therefore our study can help develop optimal semi-automatic segmentation criteria for a more accurate diagnosis of IPH.

There were a few limitations in this study. First of all, there was no histological confirmation since our study did not include patients who underwent carotid endarterectomy. Therefore we did not have histologic validation of the presence, absence, and size of IPHs identified on MRI images. Secondly, a relatively small number of patients were included in our study.

## 5. Conclusions

The ICC value of manual vs. 150% criteria and manual vs. 175% criteria are more reliable for quantification of IPH volume than that of manual vs. 200% criteria. Semi-automatic classification tools may be beneficial in large-scale multicenter studies to reduce image analysis time and avoid bias between human reviewers.

## Figures and Tables

**Figure 1 diagnostics-09-00184-f001:**
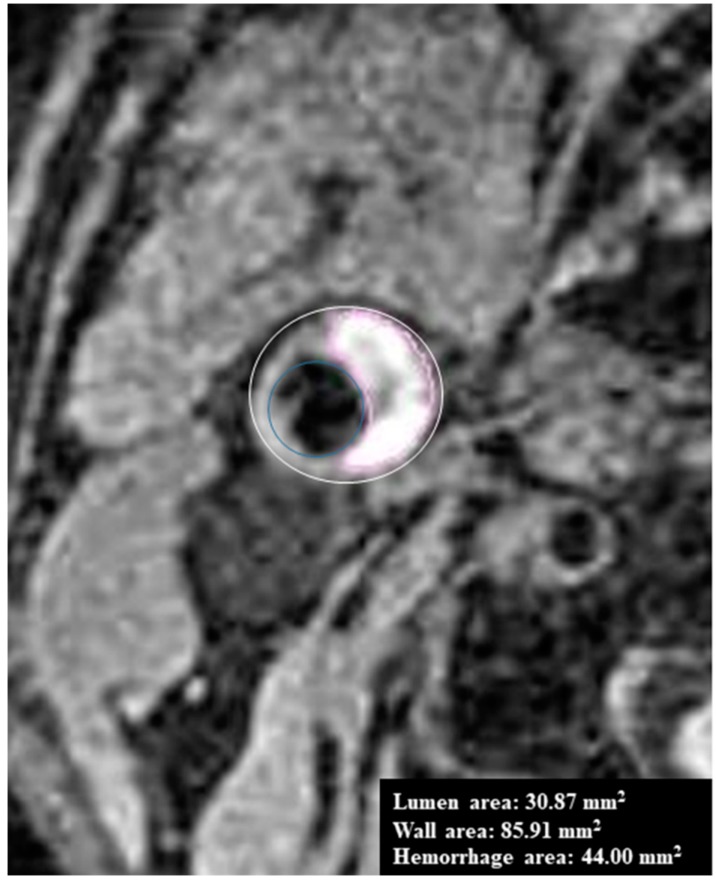
Intraplaque hemorrhage (IPH) manual segmentation using MRI-PlaqueView.

**Figure 2 diagnostics-09-00184-f002:**
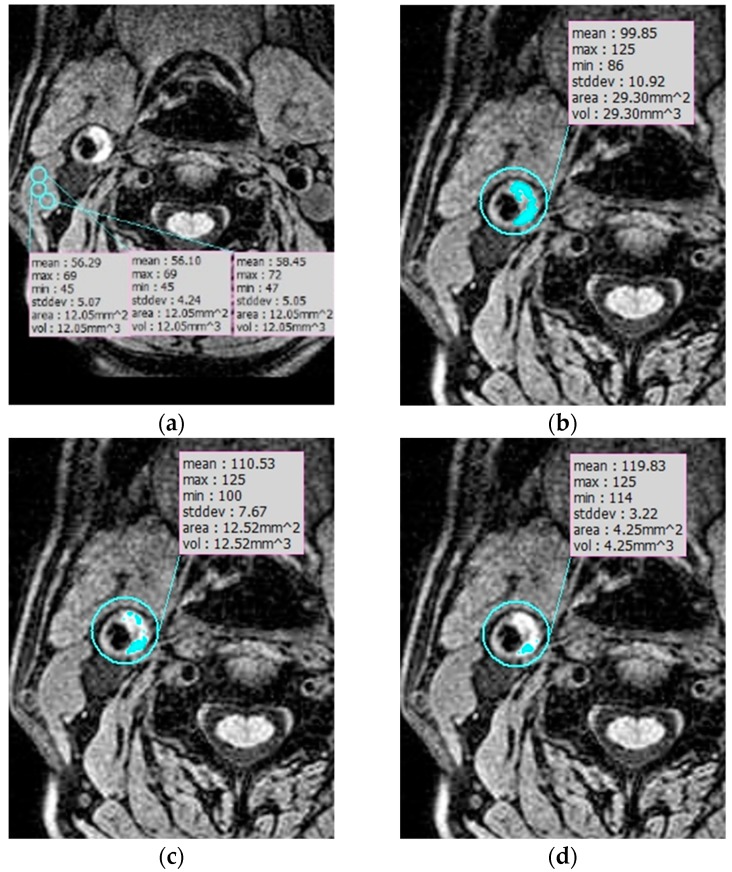
IPH semiautomatic segmentation. (**a**) Measuring the signal intensity of three points in the sternocleidomastoid muscle (SCM) to calculate the mean signal intensity of SCM. (**b**–**d**) IPH detection and volume were measured by the area that showed signal intensity of >150% (**b**), >175% (**c**), and >200% (**d**) of the mean intensity of the SCM.

**Figure 3 diagnostics-09-00184-f003:**
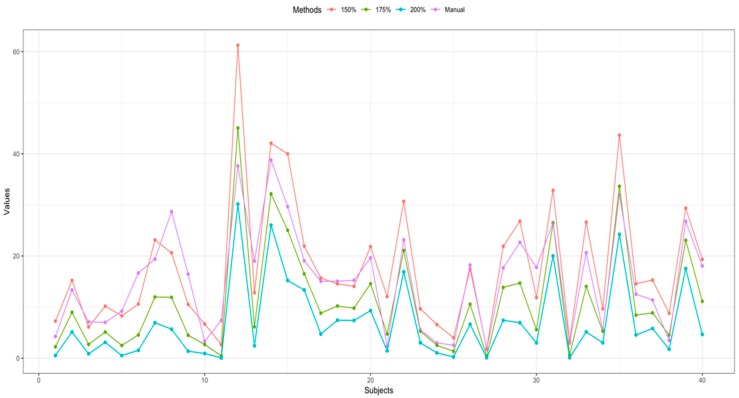
Comparison of IPH volume between manual segmentation and semiautomatic segmentation of each patient.

**Table 1 diagnostics-09-00184-t001:** Intraclass correlation coefficient between manual segmentation and semiautomatic segmentation for detection of intraplaque hemorrhage.

	ICC	95% CI
Manual—150% ($\rho_{1}$)	0.861	(0.754, 0.924)
Manual—175% ($\rho_{2}$)	0.809	(0.669, 0.894)
Manual—200% ($\rho_{3}$)	0.491	(0.218, 0.693)

Note. ICC = intraclass correlation coefficient, CI = confidence interval.

**Table 2 diagnostics-09-00184-t002:** Statistical analysis of intraclass correlation coefficient between manual segmentation and semiautomatic segmentation for detection of intraplaque hemorrhage.

Hypothesis	Fisher’s Test Statistic	Adj. *p*-Value (Bonferroni)
$H_{0}:\;\rho_{1}=\rho_{3}\;vs\;H_{A}:\;\rho_{1}>\;\rho_{3}$	4.4686	0
$H_{0}:\;\rho_{2}=\rho_{3}\;vs\;H_{A}:\;\rho_{2}>\;\rho_{3}$	3.3755	0.0012

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
