# Peer review of "Quantification of Carotid Intraplaque Hemorrhage: Comparison between Manual Segmentation and Semi-Automatic Segmentation on Magnetization-Prepared Rapid Acquisition with Gradient-Echo Sequences"

_diagnostics, 2019, doi:10.3390/diagnostics9040184_

Round 1

Reviewer 1 Report

In figure 1, the lumen area, wall area and hemorrhage area are listed but the actual segmentation only shows one region. Please provide segmentation for all three areas and/or explain this figure better. 

Author Response

In figure 1, the lumen area, wall area and hemorrhage area are listed but the actual segmentation only shows one region. Please provide segmentation for all three areas and/or explain this figure better. 

- Answer) Yes, we inserted the line of wall and lumen segmentation in Figure 1.

Reviewer 2 Report

The paper compares manual segmentation and semi-automatic segmentation for quantification of carotid intraplaque emorrhage on widely available magnetization-prepared rapid acquisition with gradient-echo (MPRAGE) sequences.

It is  a well drawn study but i think you could be more precise about your results, especially in the Discussion section.

Citation is needed when you say (line 60) that “Manual segmentation can help acquire diverse quantitative measurements on plaque composition, but it can be time-consuming and contains bias between human reviewers due to different levels of experience especially in large-scale multicenter studies”.

In the discussion you say that your study demostrates a comparison (line 136): which one? 

You should also discuss here your data and comment them by citing literature. In the present paper the discussion is just about a review of the literature without any comparison or correlation/comment to your data/results, exception made for the final comparison with Liu et al.

Author Response

Rev#2-1) Citation is needed when you say (line 60) that “Manual segmentation can help acquire diverse quantitative measurements on plaque composition, but it can be time-consuming and contains bias between human reviewers due to different levels of experience especially in large-scale multicenter studies”.

Answer – This sentence is assumed of our study. So we made some changes to this sentence.

“We thought that manual segmentation for analyzing plaque component can be time-consuming and contains bias between human reviewers due to different levels of experience especially in large-scale multicenter studies. “

Rev#2-2) In the discussion you say that your study demostrates a comparison (line 136): which one? 

Answer) We deleted this sentence because it is not necessary in our paper.

Rev#2-3) You should also discuss here your data and comment them by citing literature. In the present paper the discussion is just about a review of the literature without any comparison or correlation/comment to your data/results, exception made for the final comparison with Liu et al.

Answer – The number of studies related to quantification of carotid intraplaque hemorrhage is limited because most studies deal with the existence of IPH.
Therefore there weren’t many papers to compare with our study. It is our desire to compare and correlate our study to other similar papers but because of this reason, there were limitations. We hope this factor is taken into consideration.